# Brain–Gut–Microbiome Interactions and Intermittent Fasting in Obesity

**DOI:** 10.3390/nu13020584

**Published:** 2021-02-10

**Authors:** Juliette Frank, Arpana Gupta, Vadim Osadchiy, Emeran A. Mayer

**Affiliations:** G. Oppenheimer Family Center for Neurobiology of Stress and Resilience, Vatche and Tamar Manoukian Division of Digestive Diseases, David Geffen School of Medicine at UCLA, Los Angeles, CA 90095-7378, USA; jfrank19@g.ucla.edu (J.F.); AGupta@mednet.ucla.edu (A.G.); VOsadchiy@mednet.ucla.ed (V.O.)

**Keywords:** ingestive behavior, food addiction, gut microbiome, diurnal rhythm, weight loss

## Abstract

The obesity epidemic and its metabolic consequences are a major public health problem both in the USA and globally. While the underlying causes are multifactorial, dysregulations within the brain–gut–microbiome (BGM) system play a central role. Normal eating behavior is coordinated by the tightly regulated balance between intestinal, extraintestinal and central homeostatic and hedonic mechanisms, resulting in stable body weight. The ubiquitous availability and marketing of inexpensive, highly palatable and calorie-dense food has played a crucial role in shifting this balance towards hedonic eating through both central (disruptions in dopaminergic signaling) and intestinal (vagal afferent function, metabolic toxemia, systemic immune activation, changes to gut microbiome and metabolome) mechanisms. The balance between homeostatic and hedonic eating behaviors is not only influenced by the amount and composition of the diet, but also by the timing and rhythmicity of food ingestion. Circadian rhythmicity affects both eating behavior and multiple gut functions, as well as the composition and interactions of the microbiome with the gut. Profound preclinical effects of intermittent fasting and time restricted eating on the gut microbiome and on host metabolism, mostly demonstrated in animal models and in a limited number of controlled human trials, have been reported. In this Review, we will discuss the effects of time-restricted eating on the BGM and review the promising effects of this eating pattern in obesity treatment.

## 1. Introduction

Currently, one in three Americans of all ages—over 100 million people—have obesity. Obesity is defined as a body mass index (BMI) ≥30 kg/m^2^ and a BMI ≥40 kg/m^2^ is considered extreme obesity, with overweight classified as 25–29.9 kg/m [1]. As obesity rates hit peak levels, causing a major public health crisis, many Americans are looking to popular diet programs for a simple solution. There are countless diet recommendations for losing weight and reducing cardiovascular risk factors associated with being overweight such as heart disease, metabolic syndrome, high blood pressure, high cholesterol and C reactive protein (inflammation in the body). The great majority of diets aim to either restrict or reduce the total or relative amounts of macronutrients, e.g., fat, protein and carbohydrates, without much consideration of the role of the gut’s microbial ecosystem in modulating the brain–gut interactions. Although many Americans are looking for a simple solution for weight loss and decreasing their cardiovascular risk factors, most diet programs, which have recently increased in popularity, require a reduction in overall dietary intake. While effective in the short term, most of these diets involving either daily or intermittent fasting have not proven to be a long-term, sustainable solution [2]. More recently, changes in the timing of food intake, without caloric restriction, so called time-restricted eating (TRE), have been proposed as more effective long-term strategies to combat obesity and its metabolic complications [3]. TRE allows a combination of a ketogenic state during the 16–18 h without food intake (including the time of sleep) and the benefits of a healthy, largely plant based diet during the 6–8 h eating period that shows promising early effects as a sustainable strategy for maintaining optimal metabolic function. However, while numerous preclinical studies have shown the effects of (TRE) on the gut microbiome and on body weight, there is not sufficient clinical evidence to date to evaluate their long-term clinical effectiveness [4].

## 2. The Role of Altered Brain–Gut–Microbiome Interactions in Obesity

Obesity is a complex problem that requires a systems biological-based approach in order to understand its pathophysiology and to find effective long-term solutions and prevention strategies. There is an expanding body of preclinical studies that show support for alterations in bidirectional signaling within the brain–gut–microbiome (BGM) system in the pathophysiology of obesity mediated by metabolic, endocrine, neural and immune system-mediated mechanisms [5]. For example, specialized cells such as enteroendocrine and enterochromaffin cells in the gut, which can sense many of the metabolites (such as short-chain fatty acids) produced by the gut microbiota from ingested dietary fiber (better referred to as Microbiota Accessible Carbohydrates, or MACs [6]), send signals to the brain either via the bloodstream or via vagal afferent pathways. Gut to brain signaling can also occur via gut microbes interacting with gut-based immune cells. These interactions can either result in local effects in the gut, effects on vagal afferent terminals or lead to systemic immune activation (metabolic toxemia), which can ultimately affect other organs and target cells in the body, including glial cells in the brain, leading to neuroinflammation [7,8]. On the other hand, important signaling pathways from the brain that can modulate the gut and its microbiota include the autonomic nervous system (ANS), which is responsible for regulating gastrointestinal processes such as immune activation, intestinal permeability, gut microbial abundances and microbial gene expression patterns in response to internal or external perturbations [9,10,11]. These diverse bidirectional signaling mechanisms between the brain and the gut allow for extensive communication within the BGM system, ensuring high adaptability to different dietary patterns and to different emotional and environmental states. As expected from any complex system, the existence of bidirectionality in BGM interactions involving multiple feedback loops is the basis for regular oscillatory variations in health, and the breakdown of this rhythmicity is a characteristic indicator for metabolic disturbances [12,13,14,15]; see Figure 1.

BGM interactions are not a static system but they go through different phases throughout the lifespan, with an early developmental phase, a phase during adulthood when it adapts to environmental factors, particularly diet, and a late phase in advanced age [16]. The most critical time for the programming of the gut microbiome and bidirectional BGM interactions is during pregnancy and the first 1000 days of life [17]. During this influential time, the basic architecture of the gut microbiome and key circuits within the brain are being established and “hardwired”. However, throughout life the individual gut microbiome is highly susceptible to both internal influences (from metabolism, gut–microbiota interactions and energy expenditure) as well as influences from the environment (including household members, pets, diet, stress, medication, infections). However, despite this plasticity, the BGM system is a highly resilient and resistant ecosystem that can withstand most perturbations.

Under normal conditions, the BGM system plays a prominent role in regulating ingestive behaviors in a way that keeps body weight stable by balancing the metabolic needs of the body with hedonic impulses from the central reward system. There is extensive literature about the connection between homeostatic food intake and body weight maintenance through interactions between hypothalamic nuclei and orexigenic and anorexigenic gut hormones as well as chemical signals generated by the gut microbiota or derived from adipose tissue, specifically leptin [5]. The complex balance between orexigenic (ghrelin, insulin) and anorexigenic signals including cholecystokinin, neuropeptide Y and glucagon-like peptide-1, gut microbial metabolites (short chain fatty acids (SCFAs) and amino acid metabolites), anorexigenic signals from adipose tissue (leptin), stress mediators (corticotropin releasing factor, norepinephrine, cortisol), the central reward system and prefrontal cortical inhibitory mechanisms, ultimately determines the timing and the quantity of food that we eat. A biasing of this tightly regulated system by a reduction in anorexigenic and inhibitory signals resulting in a dominance of hedonic impulses has been implicated in the development of predominantly hedonic eating behaviors, referred to as food addiction manifesting as recurrent cravings and eating beyond a person’s metabolic needs [5,18,19]. 

Despite the complexity of the BGM system, reductionist approaches to understand and treat dysregulated ingestive behavior and obesity continue to dominate much of obesity research and therapeutic approaches. For example, the most widely studied and discussed aspects of food intake in the development of obesity are the total quantity and the relative macronutrient distribution of consumed food [20]. It is only recently that the temporal dimension of food intake has received widespread scientific attention, with the realization that it plays an important role in the regulation of the gut microbiome, and of the bidirectional interaction between the brain and the gut. Similar to any system relying on feedback mechanisms, the BGM system has been found to display oscillations, which reflect the circadian rhythm and can be manipulated by intermittent fasting periods and similar dietary strategies (Figure 1).

## 3. Environmental Factors That Influence BGM Interactions in Obesity

Cheap, highly processed and easily accessible high caloric and palatable foods are abundant in the developed world [23]. Studies have shown that foods enhanced for taste and salience not only increase cravings and ingestion of these foods, but contextual factors such as stress can serve as conditional cues for future food intake and long-term weight gain [24,25,26]. In a large cross-sectional study, individuals with obesity reported an increased preference and craving for foods rich in fat and sugar [27]. In fact, overconsumption of highly palatable foods, particularly those containing high levels of fat and sugar, progressively reduces the rewarding thresholds of such foods when ingested, a situation reflecting reduced levels of dopamine and dopamine receptors in the brain. According to this dopamine deficiency hypothesis [28], an affected individual requires an increased intake of such foods to generate the same satisfaction [29,30]. While the adult microbiome is relatively resistant to short-term changes in diet [31,32], long-term consumption of highly processed foods by pregnant women has been shown to alter gut microbial diversity and relative abundances in the newborn, and exposure of infants to such foods can influence appetite preferences and eating habits that persist throughout life [33]. 

The marketing tactics of the US food industry, targeting the hedonic component of the BGM system, have played a major role in creating and maintaining the unhealthy eating habits of the majority of Americans [34]. The American food industry, including large conglomerates responsible for the production, processing, and marketing of food, follows an exclusive shareholder value-driven business model designed to sell products, without the consumers’ health being a top priority. In order to drive consumption, processed foods are high in fat and sugar, two macronutrients contributing to food addiction. Unfortunately, the U.S. government’s nutrition policy is influenced by massive lobbying efforts by these powerful corporations. This situation has allowed for many dietary guidelines and nutrition recommendations to be skewed and meddled with in order for companies to program children’s taste preferences early on by selling sugary breakfast cereals and other highly processed food items for decades [35]. The health of the American population has seen a major decline, with continuous increases in non-communicable diseases related to obesity and metabolic health during the last 75 years, including non-alcoholic fatty liver disease, cardiovascular diseases, degenerative brain disorders and even some forms of cancer [35]. An industry that markets ultra-processed, high-sugar products and targets them directly to children is bound to see negative health consequences amongst their adult consumers. Added value and convenience are driving forces in selling products in the food industry; this includes sugary cereals with added nutrients claiming to be a “balanced breakfast” and fast-food giants marketing their high-fat products as cheap, on-the-go meals.

In addition to the marketing of obesogenic foods, studies have shown that portion sizes are directly related to a compromised ability to control food intake, an important feature of food addiction and obesity [36]. Food labels and plate and utensil sizes can moderate the portion control effect by increasing food intake. The profit-driven marketing practices of the food industry have influenced the norms of Western society’s eating habits by driving large portion sizes and highly processed foods, which continues to be a major contributor to the rising rates of obesity and metabolic disease [34]. 

There is growing consensus that largely plant-based diets, such as the traditional Mediterranean diet, the Okinawan diet and many traditional diets around the world, are associated with healthy aging and longevity and a lower prevalence of obesity, metabolic syndrome, cognitive decline and most chronic non-communicable diseases [20]. These health-associated diets contain primarily plant-based foods, including tubers, beans, seeds, nuts, olive oil and a moderate amount of fish and poultry. The most important health benefits from these diets are related to the high content and variety of polyphenols (often wrongly referred to as antioxidants) and plant-derived fiber, or MACs [6]. Both of these plant-derived diet components are primarily metabolized and absorbed in the distal small intestine and colon, as they cannot be absorbed in the small intestine and require the gut microbiome for metabolism. It does not hurt that the diet is delicious and is generally consumed in social settings, which highly contributes to its success. There are hundreds of different fiber types [37] and several thousand polyphenols and phenolic compounds found [38] in plants, with the majority belonging to the flavonoid family. Polyphenols are found in the leaves, fruits, seeds and roots of most plants and act as their own medicine to help them resist disease, pests, drought and other harmful influences. It has been demonstrated that the variety of such plant-based components, e.g., MACs and polyphenols plays an important role in the diversity and richness of the gut microbiome [39]. 

### 3.1. The Effect of Psychosocial Stress on BGM Interactions in Obesity

It is common for people experiencing high and or chronic stress to have an increase in appetite and cravings for “comfort foods”, which are generally foods high in sugar, fat and salt [19,40,41]. The resulting change in eating behaviors often leads to weight gain and metabolic syndrome. Specifically, for those who fall into the obese range, there is a high correlation between stress and abnormal, usually unhealthy, eating behaviors such as cravings and snacking. For example, a study of 339 adults (mean BMI = 26.7 ± 5.4 kg/m^2^) showed that chronic stress can influence levels of the orexigenic hormone insulin, as well as glucose and cortisol responses, which oftentimes leads to cravings and an increase in food intake and weight gain [42]. Although the consumption of “comfort foods”, which generally occurs after dinner and often late at night can cause an immediate feeling of perceived relief from stress, many studies have shown that ingestion of highly palatable foods actually leads to increased autonomic responses, an increase in cortisol and ghrelin levels, and can disrupt the hypothalamic pituitary adrenal (HPA) axis, which have all been associated with a surge in cravings and unhealthy eating habits [41,43,44,45]. Similar to sugary foods, salty and fatty comfort foods have high addictive potential.

In several preclinical models, it has been shown that persistent and variable psychosocial stress leads to a reduction in the richness and diversity in gut microbiota. In addition, a reduction in the relative abundance of several microbial taxa was observed, including Lactobacilli [46] and Akkermansia [47], microbes which have been shown to have advantageous effects in reducing the risk of obesity and other metabolic diseases [48]. These stress-induced disruptions were tied to shifts in the functional profile of the gut microbiome and gut enzymes including altered metabolism of SCFAs, tyrosine and tryptophan [47].

### 3.2. The Effect of Gut Microbial Immune Activation on BGM Interactions in Obesity

Mouse models of obesity have shown that mice fed a high-fat diet (60% lard) very low in dietary fiber have experienced long-term disruptions in gut microbiota diversity [6], while high-fiber diets resulted in positive alterations in ingestive behavior, such as decreased food intake and increased satiety [39]. Limited fiber intake resulted in a substantial reduction in diversity and abundance of microbiota, which was largely reversible within a single generation when animals were given a high-fiber diet. However, a continuation of the low-fiber diet over multiple generations resulted in a permanent loss of microbial diversity, which was not recoverable even with an increased fiber supplementation [6]. Low levels of fiber in the diet cause gut microorganisms such as certain *Akkermansia muciniphila* strains to consume the glycans making up mucins in the mucus layer of the gut, which compromises the intestinal barrier function and leads to a condition referred to as “leaky gut” [6]. Reduced intestinal barrier function can result in increased access of membrane-bound lipopolysaccharide (LPS) from Gram-positive microorganisms to TLR4 receptors on host epithelial and immune cells, contributing to inappropriate activation of the intestinal immune system [49,50,51,52]. When inflammatory mediators spread beyond the gut, systemic immune activation involving multiple organs, including the brain can ensue, a state that has been referred to as metabolic endotoxemia [51].

This metabolic endotoxemia state has shown to decrease central satiety mechanisms by influencing enteroendocrine cell secretion of the satiety hormones PYY, cholecystokinin and 5-HT [53,54,55], as well as reducing the expression of anorexigenic leptin receptors and peptide receptors on vagal afferents [56] and in the hypothalamus [57]. In the presence of a high-fat diet, vagal afferent neurons remain in an orexigenic state no matter whether food was ingested, which causes hyperphagia and obesity [57]. Along with shifts in ingestive behavior, there are likely various other mechanisms driving high-fat-diet-induced obesity, including remodeling of the intestinal transcriptome caused by gut microbiota in order to favor an obesogenic signaling cascade [58].

## 4. Circadian Variations in BGM Interactions

The time of day plays an important role in metabolism and energetics, as well as most physiological functions, such as patterns of hormone secretion, physical coordination and sleep schedules. Much of mammalian behavior and physiology functions are organized around circadian rhythms including ingestive behavior [59] and activity of the digestive system such as motility and secretory patterns [59]. The suprachiasmatic nucleus (SCN), a tiny region of the brain in the hypothalamus located directly above the optic chiasm, is considered the “master clock”, responsible for pacing and synchronizing all circadian rhythms in mammals [59,60]. Disruption of the circadian rhythm can affect gastrointestinal function, metabolism and ingestive behaviors, and considerable preclinical and clinical evidence has demonstrated a higher risk of chronic diseases, particularly metabolic syndrome, obesity, chronic cardiovascular disease and cancer, when such rhythms are compromised [61]. In developed countries, the omnipresent easy availability of food at all hours of the day, long-distance air travel crossing different time zones and shift work schedules have resulted in major disruptions of the circadian rhythmicity in terms of food intake. Specifically, the expansion of the window of time during which food is consumed, including nighttime snacking, has resulted in perturbations of the circadian rhythm of BGM interactions, contributing to compromised metabolic function [60,62]. 

The gut microbiota itself follows diurnal oscillations in composition and function whose regulation are driven by host feeding rhythms. A 2014 study by Thaiss et al. demonstrated evidence for the presence of time-of-day and feeding-time-specific profiles of microbiota functionality in laboratory mice [59]. Based on findings in animal models, the intestinal microbiota plays an important modulatory role in the timing of energy consumption and the functions of epigenetic and transcriptional systems [59]. Microbial functions surrounding energy metabolism, DNA repair and cell growth in these mouse models are performed during the dark (waking) phase, whereas the light phase showed more activity in “maintenance” pathways dealing with detoxification, motility and environmental sensing [59]. The results of their study exposed how the microbiota’s composition and functions fluctuate within hours consistent with the 24-h rhythmicity hypothesis of gut microbial function that results in robust oscillations and time-of-day-specific arrangements. Since the host circadian clock drives the rhythmicity of many functions, such as timing of food consumption, the timing of food intake is vital in dictating daily oscillations in microbiota composition. The results of the Thaiss study also showed that microbiota rhythmicity is a flexible process that is controlled by individual eating habits, further demonstrating that timing of food intake links the circadian patterns of host behavior to diurnal fluctuations in microbiota composition and function [59].

A disruption of the delicate balance in circadian clock networks associated with the mismatch of the timing of food intake and the circadian rhythm can lead to dysbiosis, or microbial maladaptation, which can contribute to the development of obesity and other metabolic malfunctions [22]. Disruption of the balanced gut microbiome not only overrides the host’s usual chromatin and transcriptional oscillations, but it also resets oscillations throughout the entire genome as well as those in both the intestine and liver [59]. 

## 5. Clinical Implications

As obesity and related metabolic diseases continue to remain a major public health issue in the U.S. and globally, there is a long list of recommended approaches for losing and controlling weight. While several of these strategies have shown short-term (up to 12 months) effectiveness in weight reduction and improvement in metabolic dysfunction, the majority of interventions have turned out to be largely ineffective in the long term [20]. There are countless diet recommendations for losing weight and reducing cardiovascular risk factors associated with being overweight. Most of these diets require restrictions or redistribution of macronutrients such as fat, protein and carbohydrates, without acknowledging the downstream effect of such diets on the gut microbiome, and the role of the microbiome in the development of resilience against weight loss [59]. Most studied diets result in substantial short-term improvements in cardiovascular risk factors, particularly blood pressure, and modest reduction in weight loss at the six-month mark, but by the 12 month follow-up, the effects largely disappeared for all popular diet programs [20]. Most of the behavioral shifts initially recommended to combat obesity, such as limited caloric intake and increased exercise, have not been effective long-term methods in most obese individuals. This is mostly due to strict limitation diets and other unsustainable methods, which reduce long-term compliance with such lifestyle changes. 

## 6. Bariatric Surgery

Considering that many of the known signaling molecules regulating hunger and appetite originate from enteroendocrine cells of the gut, it is traditionally assumed that the gut is the primary determinant of a person’s eating habits. Unsurprisingly, many treatments for obesity, such as various types of bariatric surgery, have been focused on altering the way the upper GI tract processes and assimilates food. 

Bariatric surgery refers to different surgical interventions aimed at reducing gastric storage capacity as well as reductions in food addiction and overall appetite. Although it remains the only intervention achieving robust long-term weight reduction, even in morbidly obese patients, the invasiveness, cost and common side effects preclude it as a widespread therapeutic option [63,64,65]. These techniques and clinical results have been reviewed extensively elsewhere [5]. Preliminary research in obese subjects undergoing bariatric surgery have demonstrated that the observed immediate metabolic improvements are not related to a mechanical interference with gastric capacity and secondary reduction in ingestive behavior and in absorption, as originally thought, but involve alterations of communication within the BGM system. Preliminary studies in individuals undergoing bariatric surgery suggest that the changes that occur in the gut microbiome composition and microbial metabolism of aromatic amino acids and glutamate are associated with a reduction in appetite, food addiction and alteration in food preferences, suggesting a possible relationship between these metabolites and dietary behavior [66,67,68,69,70,71]. An additional mechanism involved in the effectiveness of bariatric surgery is the significant reduction in systemic low-grade inflammation associated with this procedure [72]. It has been suggested that this anti-inflammatory effect may reestablish hypothalamic sensitivity to insulin and satiety signals, improving homeostatic regulation [73,74,75,76]. The observed dietary shift away from hedonic eating suggests a possible relationship between changes in gut microbial signaling and behavioral responses. However, more research is required to confirm this potentially causal relationship between gut microbial metabolites and food addiction in humans. Bariatric surgery is effective for individuals struggling with morbid obesity, but given its cost and undesired side effects, it is not a practical widespread therapy for losing weight or maintaining a healthy body weight.

## 7. Weight Loss Medications

Another weight loss strategy available in the U.S. is represented by several anti-obesity medications aimed at suppressing appetite [77,78,79,80]. Some of the drugs currently on the market include phentermine, bupropion, naltrexone and lorcaserin, which directly target the hypothalamus to suppress hunger signals [66]. Other medications aim to shift reward circuits in the brain to reduce the factor of enjoyment of appetizing foods and decrease the neural reaction to food cues at reward regions in the brain. These anti-obesity medications have shown limited effectiveness, making them an unreliable solution for weight loss. Their possible effects on the gut microbiota are unknown. 

## 8. Microbiome-Targeted Therapies

Microbiome-directed therapies such as treatment with novel probiotics or fecal microbiota transplantation (FMT), represent a novel therapeutic option for obesity and metabolic syndrome. Small clinical studies, including one study with 18 individuals, 9 receiving autologous FMT and 9 receiving allogenic FMT from a lean donor, and a second study which had 38 individuals, 12 receiving autologous FMT and 26 receiving allogenic FMT from a lean donor, have shown that FMT from a lean donor resulted in an increase in butyrate-producing bacteria and improved insulin sensitivity in recipients with metabolic syndrome [81,82]. However, the improved insulin sensitivity and associated changes in the fecal microbiome were not sustained at 18-week follow-up, consistent with the resilience of the individual core microbiome [82]. 

It is commonly known that microbial products such as SCFAs regulate feeding behavior in animal models via central mechanisms. For example, consumption of a specific type of fiber that selectively increases gut microbial propionate production was correlated with decreasing the temptation and brain reward signaling to highly palatable food images in a study with 20 healthy, non obese men [83]. Methods involving diet-induced alterations of the gut microbiome have been shown to be effective in the short term, without long-term sustainability [2,82]. However, for these treatments to be effective, sustainable solutions, there must be continued dietary and behavioral changes that feed the gut microbiota the necessary nutrients to continue the positive changes.

## 9. Temporal Restriction of Food Intake 

Intermittent fasting (IF) is used is an umbrella term referring to various fasting methods restricting caloric intake, such as alternate-day fasting (ADF) and time-restricted eating (TRE), even though the latter represents a fundamentally different approach. 

As stated before, many recent studies have shown that metabolic pathways, including pharmacological targets, have diurnal rhythms. It is hypothesized that under normal healthy conditions, the cyclical expression of metabolic regulators coordinates a wide range of cellular processes for more efficient metabolism [22]. There has been an increase in interest in the potential benefits of various types of IF for the treatment of diseases related to metabolic disturbances, such as obesity, cardiovascular health and ageing. IF is a method of fasting that requires a reduction in daily caloric consumption for various periods of time. There is extensive literature of preclinical studies which has shown that IF can induce longevity and improve metabolic health [3,84]. These studies in laboratory animals have demonstrated that IF can lead to an improvement in metabolic function and aid in controlling hormonal changes, inflammatory reactions, lipid metabolism, and insulin sensitivity. ADF requires 24-h periods of no or minimal calorie consumption (25% of energy needs) alternating with 24-h periods of normal energy consumption. Another popular IF method is the 5:2 eating pattern that consists of 2 nonconsecutive days of fasting and 5 days of unrestricted eating in a 7-day cycle [85]. IF is not a newfound practice; it has deep historical roots and has been used in many religions for spiritual or physical benefits [86,87]. When viewed together, these positive effects support a causal relationship between fasting and reduced prevalence of cardiovascular and neurodegenerative diseases, obesity, type II diabetes and cancer [88]. Although some animal studies show promising effects of IF, the long-term effectiveness of this dietary method, which requires calorie restriction, has been questioned, and is primarily limited by compliance [87].

## 10. Time-Restricted Eating

Time-restricted eating (TRE) is another proposed therapeutic approach for obesity and metabolic dysregulation which has gained significant recent attention [89]. Postulated health benefits of TRE are summarized in Figure 2. TRE restricts food intake to a window of 6–8 h during a 24-h period, with no calorie consumption outside of this window, as opposed to the window of 15–17 h of food intake which is common in Western countries. In principle, TRE is an attractive strategy, allowing the combination of a ketogenic state during the 16 h of noneating and a healthy, largely plant-based diet during the 8-h eating period. However, the effectiveness of such a combination of TRE with a specific diet has not been evaluated in controlled studies. TRE is a more viable option than IF since it does not require the reduction of overall caloric intake during the day, but rather to compress the window of unrestricted energy consumption within a 24-h time period, with a large part of the food restriction period taking place during sleep. A regular daily eating schedule is one of the main determining factors of diurnal rhythms in metabolic pathways. Implementing TRE where energy consumption is confined to an 8-h window prevents the harmful effects of metabolic diseases caused by high-fat, high-sugar diets without limiting daily caloric intake. 

Based largely on studies in laboratory mice, time-restricted eating is hypothesized to influence metabolic regulation through its effects on circadian biology, the gastrointestinal microbiota and modifiable lifestyle behaviors. Disruptions in these circadian clock-regulated systems can increase risks of developing obesity, cardiovascular disease, diabetes and cancer due to a hostile metabolic network. In preclinical models, diurnal oscillations in microbial metabolite production have a major effect on the circadian epigenetic and transcriptional landscape of the gut, liver and probably the brain, and eating behaviors such as unlimited access to a high-fat diet, leading to an interference with diurnal oscillations, have been associated with obesity and metabolic syndrome [60]. In animals, TRE actually reduces whole-body fat accumulation and inflammation while improving glucose tolerance, reducing insulin resistance, improving homeostasis, and restoring cholesterol homeostasis [22]. Recent research in animals demonstrates the importance of normal daily circadian rhythms for maintaining optimal metabolic function and a number of publications have demonstrated the effectiveness of such diets in animal models of obesity and metabolic diseases [3,62,90]. Two recent studies, have looked at the effects of TRF in mice with several metabolic dysregulations related to obesity, diabetes, insulin resistance, inflammation and high cholesterol [22,62]. In both studies, TRF was shown to greatly reduce the impact on normal metabolic function stemming from an obesogenic diet that was sustained throughout the duration of the TRF regimen [3]. Chaix et al. [22] study tested TRF (8–9 h food access in the active, dark phase) on mice with preexisting conditions of obesity and type II diabetes. The TRF regimen showed stabilizing effects and a reversal of the progression of metabolic diseases. TRF’s beneficial effects were even maintained on weekends when the fasting regimen was briefly paused due to ad libitum eating. The second study looking at obese mice with preexisting conditions showed that ad libitum exposure to a high-fat diet led to shifts in circadian rhythms and feeding habits that caused an increase in energy intake and weight gain [84] which could be reversed by TRF [3,62]. 

Allowing the digestive tract time periods without food forces the body to burn fat during these times instead of using energy from a continuous glucose supply. In the absence of food ingestion, a metabolic change occurs which forces the liver to produce ketones from the metabolism of body fat when glucose is inaccessible. As well as acting as fuel, ketone bodies are signaling molecules that have significant effects on major functions of the cells and organs, including the brain. These systemic and cellular responses switched on during fasting are thought to stay activated and fortify mental and physical functioning as well as disease resistance even after resumption of food ingestion. Other potential factors that might contribute to the benefits of TRE include the subconscious reduction of snacks and overall calories and changes in the gut microbial environment due to an increase in fasting behaviors of motility and secretion [90]. 

Despite impressive results in rodent models, to date, well designed clinical trial results to determine the effectiveness of this temporally restricted food intake without calorie restriction or change in diet in obese subjects with metabolic disturbances are limited and inconsistent. One study with overweight but otherwise healthy individuals adhering to TRE significantly reduced their daily caloric consumption mainly by eliminating late-night alcohol and snacks, which resulted in sustained weight loss for up to one year [91]. The TREAT randomized clinical trial published in JAMA Internal Medicine in 2020 tested the effects of TRE on weight loss and other metabolic parameters in obese women and men (participants had a mean weight of 99.2 kg and mean BMI of 32.7). The TRE group followed a 16:8 h eating schedule and was instructed to eat ad libitum from 12 pm until 8 pm and completely abstain from caloric intake from 8 pm until 12 pm the following day, while the consistent meal timing group (CMT) was instructed to eat three structured meals per day. The investigators hypothesized that since TRE does not require a decrease in 24-h caloric intake, it may affect energy expenditure to achieve a negative calorie balance. The primary outcome of TRE in the trial was significant weight loss amongst the TRE group and little to no change in weight within the CMT group. They also found a significant difference in appendicular lean mass index between groups but no changes in any other secondary outcomes such as fasting insulin, fasting glucose or estimated energy intake. The study concluded that without other interventions, TRE is not necessarily more effective for weight loss than eating throughout the day and may cause loss of lean muscle mass [4]. Previous small studies with time-restricted eating in overweight or obese humans showed a reduction in overall calorie consumption which led to decreased bodyweight and fat mass [4]. According to the results of the TREAT study, adherence to a regular physical exercise program and adherence to a healthy diet may be able to prevent such unexpected side effects. More research on humans is needed to make more definite conclusions about TRE, including the possible combination with dietary changes which may be a more viable long-term effective weight loss strategy.

## 11. Summary and Conclusions

Complex bidirectional interactions within the BGM system play a crucial role in the regulation of ingestive behavior. Several dietary and environmental factors have been identified that can alter the tightly regulated communication within the BGM system, shifting the balance from homeostatic to hedonic food intake, as well as cause various kinds of altered ingestive behaviors, such as increased cravings and binge eating and resulting in metabolic disturbances. Based largely on results from preclinical studies, the temporal pattern of food intake, in addition to type, quantity and quality of consumed food, plays an important role in obesity and associated metabolic consequences. Several strategies related to the timing of food intake have been evaluated, primarily in rodent models. While most intermittent fasting strategies have not been found to be superior to conventional dieting in achieving sustained, long-term weight loss and control, TRE, particularly when combined with a healthy, largely plant-based diet, may be a more practical approach with higher compliance and with long-term benefits. More confirmatory human studies, including more in depth evaluation of the role in gut microbial function of the beneficial TRE effects are needed to firmly establish this intervention as a cost-effective therapy for obesity.

## Figures and Tables

**Figure 1 nutrients-13-00584-f001:**
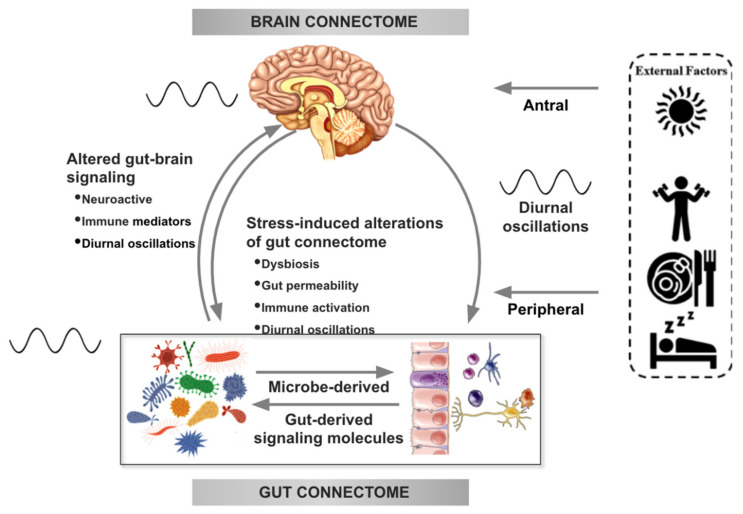
Bidirectional interactions within the brain gut microbiome (BGM) system are modulated by diurnal oscillations. Outputs generated by the central autonomic brain network modulate the activity of various components of the gut connectome (immune, endocrine, epithelial cells, gap junctions, smooth muscle, neurons) which influence gut function and interactions of the gut with the microbiome. Sympathetic nervous system outflow from the brain can also modulate gut microbial gene expression and function directly. Both gut and brain induced modulation of gut microbial metabolites and secretion of neuroactive neuroactive and inflammatory signaling molecules feed back to the brain and modulate the activity of brain networks. Diurnal variations (depicted by an oscillating line), food intake, sleep and physical activity modulate autonomic nervous system (ANS) activity as well as gut microbial composition and function. Oscillations between the brain and the gut are synchronized with the central clock in the hypothalamic suprachiasmatic nucleus. Modified with permission from Osadchiy et al. [21] and Chaix et al. [22].

**Figure 2 nutrients-13-00584-f002:**
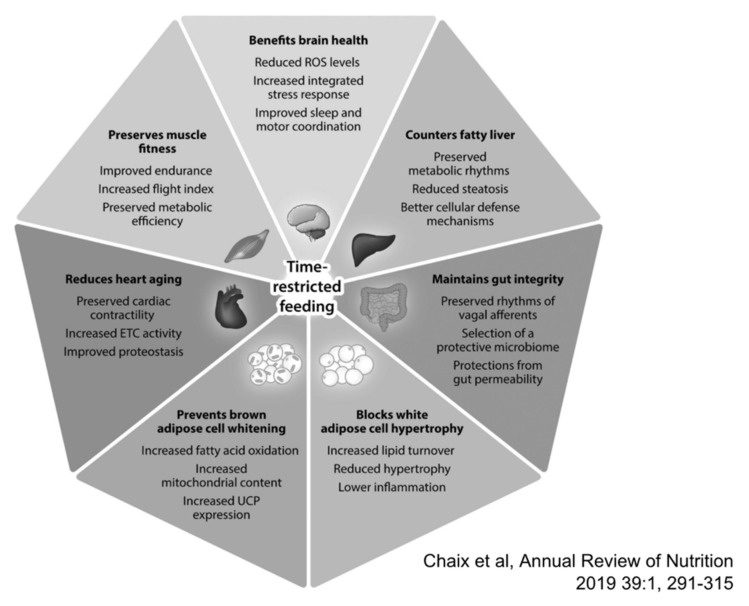
Reported benefits associated with time-restricted eating (TRE). Based largely on preclinical studies a wide range of benefits have been postulated for TRE, while clinical studies are non-conclusive to date. With permission from Chaix et al. [92].

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
