# Peer review of "Brain–Gut–Microbiome Interactions and Intermittent Fasting in Obesity"

_nutrients, 2021, doi:10.3390/nu13020584_

Round 1
Reviewer 1 Report
Review of the review manuscript entitled “Brain–Gut–Microbiome Interactions and Intermittent Fasting in Obesity” submitted by Juliette Frank, Arpana Gupta, Vadim Osadchiy and Emeran A. Mayer
In this manuscript the authors review the topics of interactions between the intestinal microbiota/gut and brain and the bidirectional signaling between them, circadian regulation of these systems and processes, how disruption of this signaling by intrinsic and extrinsic factors, including psychological stress, leads to metabolic diseases, a discussion of political and commercial influences of industrial food production and marketing on obesity and metabolic syndrome, a survey of surgical, pharmacological and interventional strategies for treating obesity and metabolic syndrome and their significant limitations, and finally how altering the timing food consumption might affect the microbiota and gut-brain signaling and might be clinically useful in treating various aspects of metabolic syndrome.
Overall, this review provides a useful synopsis of the literature in these areas. While the review cover quite a wide-ranging set of topics broadly related to metabolic health and disease, the high level prospective of this review is broadly valuable.
Minor critiques:
- Additional references for certain statements would be useful:
- Lines 95-98
- Lines 181-182
- Lines 239-242
- In lines 246-247, the statement “The intestinal microbiota is the key regulator in the timing of energy consumption and the functions of epigenetic and transcriptional systems.”, is a gross overstatement, and one not claimed in the original Thaiss paper. The paper definitely makes it clear that circadian regulation of gut microbiota is critical to these regulatory aspects as demonstrated using their fecal transplant model, but the phrase “the key regulator” indicates a level of necessity, and more importantly, sufficiency not demonstrate or claimed in the original paper.
- In lines 362-364, the authors make the statement about TRE that “It is an attractive strategy allowing the combination of a ketogenic state during the 16 hours of noneating and a healthy, largely plant based diet during the 8 hour eating period.” This implies that not only the timing of feeding is changed, but the composition of the diet is also changed from a high fat/low fiber animal-based diet. In the subsequent section describing the efficacy and outcomes of various studies with TRE, was there a diet change as well as changes to feeding time/duration? If so, it is critical to make this clear in those sections as well as discussing how the change in diet away from one know to adversely impact the gut microbiota and food intake would clearly confound interpretation of the TRE studies.
- In lines 410-413, the authors describe “a more recent study”, but do not appear to describe the results of the study or provide a citation.
- The figures (use with permission and minor modification from other manuscripts), do not appear to enhance or clarify the points in the text of the review significantly. In fact. Figure 2 is not mentioned in the text.
Author Response
TO: Editorial Board, Nutrients
FROM: Emeran A. Mayer, MD
RE: Review of the review manuscript entitled “Brain–Gut–Microbiome Interactions and Intermittent Fasting in Obesity” submitted by Juliette Frank, Arpana Gupta, Vadim Osadchiy and Emeran A. Mayer
We would like to thank the Reviewers and the Editorial Board for the favorable review of our manuscript. We have addressed all comments on the point by point document below, and have marked corresponding edits to the manuscript in red ink.
We hope that the Editorial Board will find the revised manuscript acceptable for publication in Nutrients.
Kind regards,
Emeran A. Mayer, M.D. for all authors
Reviewer 1
Comment:
In this manuscript the authors review the topics of interactions between the intestinal microbiota/gut and brain and the bidirectional signaling between them, circadian regulation of these systems and processes, how disruption of this signaling by intrinsic and extrinsic factors, including psychological stress, leads to metabolic diseases, a discussion of political and commercial influences of industrial food production and marketing on obesity and metabolic syndrome, a survey of surgical, pharmacological and interventional strategies for treating obesity and metabolic syndrome and their significant limitations, and finally how altering the timing food consumption might affect the microbiota and gut-brain signaling and might be clinically useful in treating various aspects of metabolic syndrome.
Overall, this review provides a useful synopsis of the literature in these areas. While the review cover quite a wide-ranging set of topics broadly related to metabolic health and disease, the high level prospective of this review is broadly valuable.
Response:
Thank you for the favorable review of our MS
Minor critiques:
- Additional references for certain statements would be useful:
- Lines 95-98
- Lines 181-182
- Lines 239-242
Response: References have been added to MS
- In lines 246-247, the statement “The intestinal microbiota is the key regulator in the timing of energy consumption and the functions of epigenetic and transcriptional systems.”, is a gross overstatement, and one not claimed in the original Thaiss paper. The paper definitely makes it clear that circadian regulation of gut microbiota is critical to these regulatory aspects as demonstrated using their fecal transplant model, but the phrase “the key regulator” indicates a level of necessity, and more importantly, sufficiency not demonstrate or claimed in the original paper.
Response: Wording has been added as suggested by Reviewer
- In lines 362-364, the authors make the statement about TRE that “It is an attractive strategy allowing the combination of a ketogenic state during the 16 hours of noneating and a healthy, largely plant based diet during the 8 hour eating period.” This implies that not only the timing of feeding is changed, but the composition of the diet is also changed from a high fat/low fiber animal-based diet. In the subsequent section describing the efficacy and outcomes of various studies with TRE, was there a diet change as well as changes to feeding time/duration? If so, it is critical to make this clear in those sections as well as discussing how the change in diet away from one know to adversely impact the gut microbiota and food intake would clearly confound interpretation of the TRE studies.
Response: Suggested clarifications have been made
- In lines 410-413, the authors describe “a more recent study”, but do not appear to describe the results of the study or provide a citation.
Response: Description of study and citation have been added.
- The figures (use with permission and minor modification from other manuscripts), do not appear to enhance or clarify the points in the text of the review significantly. In fact. Figure 2 is not mentioned in the text.
Response: We strongly feel that the addition of both figures enhances the manuscript, and are now referring to Fig. 2 in the text
Reviewer 2 Report
The authors in the Review discuss the effects of time restricted eating (TRE) on the brain-gut-microbiome (BGM) system and review the promising effects of this eating pattern in obesity treatment.
Reviewer’s points
1-The paper is well written and quite exhaustive about the topic. Dysregulation within the BGM system play a central role in several multifactorial diseases (obesity, diabetes, etc.), but they have been demonstrated and adequately supported with experimental data mostly in animal models. Similarly, the treatment with various fasting strategies by which to restore BGM, has been mainly investigated in animal models.
For these reasons and not to confuse the reader, this reviewer would suggest underlining in the abstract that most of the experimental evidences refer to animal models.
The authors are also invited to add to the references a nice recently published paper about the effect of Intermittent Fasting in alleviating diabetes-induced cognitive impairment (Zhigang Liu et al. Nature Communications; 2020,11: 855).
2-This reviewer is not aware that the causal relationship between intermittent-fasting and gut microbiota- brain axis modulation, has yet been sufficiently proven in humans, but only hypothesized. Further, many studies are based on the changes observed in the fecal microbiome after fasting. But faeces, it is well known, only partially represent the gut (small intestine-colon) microbiome, this can lead to some misjudgment. In addition, most of the reported studies indicate that the characterization of the microbiome is mainly done at the taxonomic (16S rRNA) level; however, to investigate the microbial functions (by metatrascriptomics, metabolomics) could have given more information.
Consequently,the authors should mention in their conclusions the limits of the studies carried out so far in humans and suggest future approaches to investigate which molecules and pathways could be able to transmit the variations of the microbiota to the brain.
Author Response
TO: Editorial Board, Nutrients
FROM: Emeran A. Mayer, MD
RE: Review of the review manuscript entitled “Brain–Gut–Microbiome Interactions and Intermittent Fasting in Obesity” submitted by Juliette Frank, Arpana Gupta, Vadim Osadchiy and Emeran A. Mayer
We would like to thank the Reviewers and the Editorial Board for the favorable review of our manuscript. We have addressed all comments on the point by point document below, and have marked corresponding edits to the manuscript in red ink.
We hope that the Editorial Board will find the revised manuscript acceptable for publication in Nutrients.
Kind regards,
Emeran A. Mayer, M.D. for all authors
Reviewer #2
The authors in the Review discuss the effects of time restricted eating (TRE) on the brain-gut-microbiome (BGM) system and review the promising effects of this eating pattern in obesity treatment.
Reviewer’s points
1-The paper is well written and quite exhaustive about the topic. Dysregulation within the BGM system play a central role in several multifactorial diseases (obesity, diabetes, etc.), but they have been demonstrated and adequately supported with experimental data mostly in animal models. Similarly, the treatment with various fasting strategies by which to restore BGM, has been mainly investigated in animal models.
For these reasons and not to confuse the reader, this reviewer would suggest underlining in the abstract that most of the experimental evidence refer to animal models.
Response:
We fully agree with the Reviewer’s comment and have highlighted this point in the abstract.
The authors are also invited to add to the references a nice recently published paper about the effect of Intermittent Fasting in alleviating diabetes-induced cognitive impairment (Zhigang Liu et al. Nature Communications; 2020,11: 855).
Response:
We have added statements throughout the manuscript to highlight this point. Thank you for this excellent reference.
2-This reviewer is not aware that the causal relationship between intermittent-fasting and gut microbiota- brain axis modulation, has yet been sufficiently proven in humans, but only hypothesized. Further, many studies are based on the changes observed in the fecal microbiome after fasting. But faeces, it is well known, only partially represent the gut (small intestine-colon) microbiome, this can lead to some misjudgment. In addition, most of the reported studies indicate that the characterization of the microbiome is mainly done at the taxonomic (16S rRNA) level; however, to investigate the microbial functions (by metatrascriptomics, metabolomics) could have given more information.
Consequently,the authors should mention in their conclusions the limits of the studies carried out so far in humans and suggest future approaches to investigate which molecules and pathways could be able to transmit the variations of the microbiota to the brain.
Response:
Thank you for the excellent suggestion